# Exploring the Immunomodulatory Potential of Human Milk: Aryl Hydrocarbon Receptor Activation and Its Impact on Neonatal Gut Health

**DOI:** 10.3390/nu16101531

**Published:** 2024-05-19

**Authors:** Naomi V. Wieser, Mohammed Ghiboub, Caroline Verseijden, Johannes B. van Goudoever, Anne Schoonderwoerd, Tim G. J. de Meij, Hendrik J. Niemarkt, Mark Davids, Antoine Lefèvre, Patrick Emond, Joep P. M. Derikx, Wouter J. de Jonge, Bruno Sovran

**Affiliations:** 1Tytgat Institute for Liver and Intestinal Research, Amsterdam University Medical Center, University of Amsterdam, 1105 BK Amsterdam, The Netherlands; m.ghiboub@amsterdamumc.nl (M.G.); c.verseijden@amsterdamumc.nl (C.V.); w.j.dejonge@amsterdamumc.nl (W.J.d.J.); 2Amsterdam Gastroenterology, Endocrinology, Metabolism (AGEM), 1105 AZ Amsterdam, The Netherlands; t.demeij@amsterdamumc.nl; 3Department of Pediatric Surgery, Emma Children’s Hospital, Amsterdam University Medical Center, University of Amsterdam, Meibergdreef 9, 1105 AZ Amsterdam, The Netherlands; j.derikx@amsterdamumc.nl; 4Department of Pediatrics, Emma Children’s Hospital, Dutch National Human Milk Bank, Amsterdam University Medical Center, University of Amsterdam, Meibergdreef 9, 1105 AZ Amsterdam, The Netherlands; h.vangoudoever@amsterdamumc.nl (J.B.v.G.); a.schoonderwoerd@amsterdamumc.nl (A.S.); 5Department of Pediatric Gastroenterology, Vrije Universiteit University Medical Center, 1081 HV Amsterdam, The Netherlands; 6Department of Neonatology, Maxima Medical Center, De Run 4600, 5504 DB Veldhoven, The Netherlands; hendrik.niemarkt@mmc.nl; 7Department of Electrical Engineering, Technical University Eindhoven, Groene Loper 3, 5612 AE Eindhoven, The Netherlands; 8Department of Experimental Vascular Medicine, Amsterdam University Medical Center, University of Amsterdam, 1105 AZ Amsterdam, The Netherlands; m.davids@amsterdamumc.nl; 9UMR 1253, iBrain, University of Tours, Inserm, 37044 Tours, France; antoine.lefevre@univ-tours.fr (A.L.); patrick.emond@univ-tours.fr (P.E.); 10In Vitro Nuclear Medicine Laboratory, Regional University Hospital Center of Tours University, 37044 Tours, France; 11Department of Surgery, University Hospital Bonn, 53113 Bonn, Germany; 12Emma Center for Personalized Medicine, Amsterdam University Medical Center, University of Amsterdam, 1105 AZ Amsterdam, The Netherlands

**Keywords:** prematurity, aryl hydrocarbon receptor, barrier integrity

## Abstract

Several metabolites of the essential amino acid tryptophan have emerged as key players in gut homeostasis through different cellular pathways, particularly through metabolites which can activate the aryl hydrocarbon receptor (AHR). This study aimed to map the metabolism of tryptophan in early life and investigate the effects of specific metabolites on epithelial cells and barrier integrity. Twenty-one tryptophan metabolites were measured in the feces of full-term and preterm neonates as well as in human milk and formula. The ability of specific AHR metabolites to regulate cytokine-induced IL8 expression and maintain barrier integrity was assessed in Caco2 cells and human fetal organoids (HFOs). Overall, higher concentrations of tryptophan metabolites were measured in the feces of full-term neonates compared to those of preterm ones. Within AHR metabolites, indole-3-lactic acid (ILA) was significantly higher in the feces of full-term neonates. Human milk contained different levels of several tryptophan metabolites compared to formula. Particularly, within the AHR metabolites, indole-3-sulfate (I3S) and indole-3-acetic acid (IAA) were significantly higher compared to formula. Fecal-derived ILA and milk-derived IAA were capable of reducing TNFα-induced IL8 expression in Caco2 cells and HFOs in an AHR-dependent manner. Furthermore, fecal-derived ILA and milk-derived IAA significantly reduced TNFα-induced barrier disruption in HFOs.

## 1. Introduction

Preterm birth is defined as birth prior to the completion of 37 weeks of gestation, occurring approximately in 10% of pregnancies [1]. Preterm birth is associated with an increased risk of infection, particularly early-onset sepsis and necrotizing enterocolitis (NEC) [2,3], which is associated with devastating short- and long-term consequences, such as intestinal failure [4]. Currently, the mechanisms underlying NEC and sepsis remain largely elusive; however, it is well established that a disrupted gut microbiome contributes to their emergence. Factors contributing to such observed microbiome perturbations often include a low birth weight, the frequent administration of antibiotics, and delivery via cesarean section [5,6,7]. Preterm neonates exhibit decreased gut microbiota diversity, particularly with lower levels of *Bifidobacteria* spp., and increased levels of *Enterobacter* and *Enterococcus* spp. [8]. The intestinal microbiota plays an important role in metabolizing food nutrients into multiple different metabolites, such as short-chain fatty acids (such as butyrate), secondary bile acids, or tryptophan (Trp) catabolites [9,10]. All these bacterial-produced metabolites are involved in host physiology and health [11,12]. Particularly, Trp metabolites have been shown to modulate inflammation and maintain intestinal epithelial homeostasis [10,13,14,15]. Trp can be metabolized following three routes: the indole pathway by gut microbiota and the kynurenine and serotonin pathways by mammalian (host) cells [16,17].

From the indole pathway, a set of bacteria-derived metabolites is known to activate the aryl hydrocarbon receptor (AHR), which is expressed in intestinal cells. The AHR, a receptor ubiquitously present through the body, usually resides in the cytoplasm until it is activated by the binding of specific exogenous ligands [16]. This process results in the translocation of AHR into the nucleus, where it regulates the activation of multiple downstream genes, such as *CYP1A1,* a common marker of AHR activation [18]. Both synthetic and natural ligands can activate the AHR. Tryptophan bacterial-derived metabolites are an important source of natural ligands, and new ligands are being discovered and characterized [17,19,20]. It has been shown that microbiome disruption leads to altered AHR signaling and worsened symptoms in inflammatory bowel disease (IBD) and metabolic syndrome [15,21]. AHR activation in epithelial cells has been shown to reduce inflammation and affect multiple pathways, particularly the NF-κB pathway [1,20]. However, very little is known about the role of AHR in the intestine of neonates, although recent studies relate the activation of AHR and its role on regulating inflammation to mucosal intestinal cells in preterm infants [22,23,24]. Huang et al. demonstrated that indole-3-lactic acid (ILA) could regulate IL1β-induced inflammation in multiple early-life models [22].

In neonates, the main source of Trp is milk, and it is metabolized by both host cells and the microbiome upon reaching the gastrointestinal tract. Rich in multiple bioactive nutrients, human milk plays a pivotal role in fostering the healthy development of the neonatal intestine. Human milk feeding has been shown to reduce NEC and sepsis incidence in preterm neonates [25,26].

In this study, we aimed to understand the differences in the metabolism of Trp in preterm and full-term neonatal fecal samples and human milk, as well as investigate the potential immunomodulatory potential of AHR in neonatal gut. We measured the levels of AHR-agonist metabolites in the fecal samples of full-term neonates compared to those of preterm neonates. In addition, we measured the levels of AHR-agonist metabolites in unpasteurized and pasteurized human milk and formula. Furthermore, we investigated the ability of specific AHR metabolites to regulate cytokine-induced IL8 expression and maintain barrier integrity in Caco2 cells and human fetal organoids (HFOs). We demonstrated that human milk-derived AHR agonists mediate immune regulation by reducing the expression and production of chemokine IL8 in both adult intestinal epithelial cells and human fetal organoid (HFO) models. Moreover, we provided evidence that AHR-activating indoles effectively maintain the integrity of the barrier function.

## 2. Materials and Methods

### 2.1. Human Milk and Formula

Human milk samples were obtained from a surplus of milk donated to the Dutch Human Milk Bank (*n* = 22). The formula used in this study, Nutrilon neonatal start, Nutricia (Hoofddorp, The Netherlands), was a commercially available product, prepared according to the manufacturers’ recommendations.

Both pasteurized and unpasteurized samples were stored at −20 °C. To separate the milk into skimmed components, milk was centrifuged at 4 °C for 20 min, which resulted in the dissociation of the skim and fat layers. Subsequently, the fat layer of the milk was carefully removed and stored separately at −20 °C. 

### 2.2. Neonatal Fecal Samples

Neonatal fecal samples were collected in the Amsterdam UMC and MMC Veldhoven (The Netherlands) from healthy full-term (*n* = 9) and preterm neonates (*n* = 19) at birth and after 4 weeks, as part of the eNose/NEC study. All the neonates, except for one full-term baby, were breastfed. The fecal samples were extracted from the diapers and immediately stored at −80 °C (Table 1). Study approval was provided by the local institutional review boards of all the participating centers (Amsterdam UMC and Amsterdam VUMC). Written informed consent was obtained from the parents or legal caretakers.

### 2.3. Tryptophan Metabolomics

Neonatal fecal samples (preterm *n* = 19; full-term *n* = 9) and all the layers of the divided milk (human *n* = 22; formula, *n* = 6) were processed for Trp metabolite measurement using liquid chromatography–tandem mass spectrometry (LC-MS). The Trp metabolites’ concentrations and composition from the fecal and milk samples were quantified as previously described [27]. The samples were pretreated as follows: (1) for the fecal samples, 3 mg of lyophilized fecal material was mixed with 900 μL of MeOH/H_2_O mixture (1:1), added according to internal standards; and, (2) for the milk samples, 100 μL of milk was mixed in 300 μL of methanol and 100 μL of a solution of internal standards [28].

LC-MS was performed as previously described. A calibration curve was created for each metabolite by calculating the intensity ratio obtained between the metabolite and its internal standard. These calibration curves were then utilized to determine the concentrations of each metabolite in the fecal samples. The specific calibration ranges and internal standard concentrations associated with the fragmentation parameters have been previously described [29]. 

The Trp metabolism was assessed by measuring the total metabolites involved in each pathway. The sum of the kynurenine pathway consisted of picolinic acid, 3-OH-kyurenin, quinolinic acid, kynurenine, 3-OH-anthanilic acid, kynurenic acid, and xanthurenic acid. The sum of the serotonin pathway consisted of 5-OH-tryptophan, *N*-acetyl-serotonin, 5-OH indole acetic acid, and serotonin. Finally, the sum of the indole pathway comprised tryptamine, indole-3-acetamide, indole-3-lactic acid, indole-3-aldehyde, indole-3-acetic acid, tryptophol, and indole-3-sulfate. These metabolites and their order are illustrated in Appendix A. 

The kynurenine pathway was quantified through the activity of the enzyme indoleamine 2,3-dioxygenase (IDO1), which was assessed by measurement of the kynurenine/Trp ratio [15]. The Trp metabolism was assessed by measuring the total metabolites involved in each pathway.

### 2.4. AHR Activity Measurement

A luciferase-based assay was used to assess AHR activity in response to different samples of human milk, formula, and specific indoles, as described previously [16]. Briefly, mouse kidney H1L1.1c2 cells (kindly shared by Dr. Michael Denison, University of California, Davis, CA, USA), which contained a dioxin response element-driven firefly luciferase reporter plasmid pGudLuc1.1 after the *Cyp1a1* gene (the AHR target gene), were seeded in a 96-well plate and stimulated with 10x diluted milk or the indoles for 4 h. Cell lysate was then used to measure luciferase activity using a luminometer. A synthetic AHR-activating positive-control 6-Formylindolo[3,2-*b*]carbazole (FICZ) as well as a negative control (culture medium) were used to confirm effectiveness and normalize the activity.

### 2.5. Cell Culture and Treatments

Caco2 cells, an immortalized human colon colorectal adenocarcinoma cell line (Caco2 line, passages 5–19; HTB-37; American Type Culture Collection, Manassas, VA, USA), were used as a model for the human intestine [30]. The cells were routinely maintained in Dulbecco’s Modified Eagle Medium (DMEM, Thermo Fisher Scientific, Waltham, MA, USA) supplemented with 10% heat-inactivated Fetal Calf Serum (FCS) (Life Technologies, Inc., Invitrogen, CA, USA), 100 μmg/mL penicillin and streptomycin antibiotic, and L-glutamine (Biocambrex, Verviers, Belgium). The cells were seeded in a 96-well plate and incubated for 24 h at 37 °C in a 5% carbon dioxide humidified atmosphere. The cells were then supplemented for 24 h with FICZ (3.5 nmol/mL), indole-3-acetic acid (IAA; 60 pmol/mL), indole-3-sulfate (I3S; 150 pmol/mL), or indole-3-lactic acid (ILA; 1 nM/mL) (Thermo Fisher Scientific). Finally, the cells were stimulated with tumor necrosis factor (TNFα, 50 ng/mL) (Thermo Fisher Scientific) for 4 h or kept unstimulated. The supernatant was collected and stored at −80 °C, and the cells were collected into TRIzol and stored at −80 °C (Thermo Fisher Scientific) for a subsequent protein and gene expression analysis.

### 2.6. Human Fetal Organoid Isolation, Culture, and Treatments

Human fetal intestinal tissues (gestational age 18–20 weeks) were obtained by the HIS Mouse Facility of the Amsterdam UMC from a clinic (The Netherlands). Fetal human organoids (HFO) were generated from isolated crypts issued from fetal small intestine as previously described in [31]. Briefly, after being cut open longitudinally, the intestine was cut into small pieces of ~5 mm and thoroughly washed with ice-cold phosphate-buffered saline (PBS). The tissue was then incubated in 2 mM Ethylenediaminetetraacetic acid (EDTA) diluted in PBS for 30 min at 4 °C. Next, the tissue was washed with PBS + 10% FCS (Bondinco, Alkmaar, The Netherlands), separating the dissociated cells into the supernatant, which was filtered through a cell strainer (70 μm). The isolated crypts were resuspended in Matrigel (Corning, NY, USA) and derived into organoids using a specific growth medium containing advanced DMEM supplemented with Glutamax (Invitrogen, Carlsbad, CA, USA), 1 M of Hepes, 100 U-mg/mL penicillin-streptomycin, a B27 supplement, a neurobasal supplement, EGF (Invitrogen, Carlsbad, CA, USA), *n*-Acetylcysteine (Sigma-Aldrich, Burlington, NJ, USA), Noggin, and Rspondin (home made by stably transfected HEK cells) and incubated at 37 °C, 5% CO_2_. The medium was refreshed every 2–3 days, and the organoids were passaged by mechanical disruption intervals of 6–10 days, as described previously [31].

### 2.7. Total RNA Isolation and RT-qPCR

RNA from Caco2 cells and HFOs was isolated using the bioline ISOLATE II mini kit (BIO-52073, Bioline, Waddinxveen, The Netherlands) according to the manufacturer’s protocol, and RNA quality was checked by Tapestation 2200 (Agilent Technologies Netherlands B.V., Amstelveen, The Netherlands). cDNA was synthesized with Random Hexamer primers (Promega, Leiden, The Netherlands), Oligo dT primers (Thermo Fisher Scientific), deoxyribonucleotide triphosphate (dNTPs) (Thermo Fisher Scientific), 1× RT-buffer (Thermo Fisher Scientific), Revertaid Transcriptase (Thermo Fisher Scientific), and Ribolock RNAse Inhibitor (Thermo Fisher Scientific).

Quantitative RT-qPCR was performed on a BioRad iCycler using the sensifast SYBR No-ROX Kit (GC-biotech Bio-98020, Meridian, Cincinnati, OH, USA) according to the manufacturer’s instructions. The most stable reference genes were selected from a panel of nine reference genes using GeNorm [32]. The relative expression of the genes with N0 values was obtained using LinRegPCR [33], and the values were normalized with the reference genes. Gene expression of the target genes was determined using the LinRegPCR software (version 2021.1) [33]. The primer sequences (Sigma Aldrich) for the *AHR* gene were F 5′-TTACAGGCTCTGAATGGCTT-3′ and R 5′-TTTTCTGGAGGAATCTGGTCT-3′; those for the *IL8* gene were F 5′-AAATTTGGGGTGGAAAGGTT-3′ and R 5′-TCCTGATTTCTGCAGCTCTGT-3′; those for reference genes *GAPDH* were F 5′-CCATGTTCGTCATGGGTGTG-3′ and R 5′-GGTGCTAAGCAGTTGGTGGTG-3′; and those for *CYCLOPHILLIN* were F 5′-ACGGCGAGCCCTTGG-3′ and R 5′-TTTCTGCTGTCTTTGGGACCT-3′. The primer sequence for *CYP1A1* has been previously described [15].

### 2.8. Measurement of Cytokine Production

The IL8 concentration was determined by the sandwich enzyme-linked immunosorbent assay (ELISA; R&D Systems, Abingdon, UK) according to the manufacturer’s protocol. Optical density was measured using a Synergy HT plate reader (BioTEK, Beun de Ronde, Abcoude, The Netherlands).

### 2.9. Immunofluorescence Cell Staining

Immunofluorescence staining of AHR was performed on Caco2 cells. The cells were fixed with 4% paraformaldehyde and then permeabilized with 0.1% Triton™ X (Sigma-Aldrich). The cells were blocked with 2% Bovine Serum Albumin (BSA). The cells were labelled with Mouse Anti-AHR Monoclonal Antibody (dilution: 1:50, FF3399, eBioscience™ (Product # 14-9854-82) overnight and labelled with Goat anti-Mouse IgG (H+L), Superclonal™ Recombinant Secondary Antibody, and Alexa Fluor 488. The nuclei (dilution: 1:2000; panel b: blue) were stained with SlowFade^®^ Gold Antifade Mountant with DAPI (Thermo Fisher Scientific). The images were captured at a 60× magnification and processed using ImageJ (Rasband, WS., ImageJ, U. S. National Institutes of Health, Bethesda, MD, USA, version 1.54f, 29 June 2023).

### 2.10. Measurement of Barrier Integrity

The HFOs were split into single cells and seeded at 10^5^ cells/mL onto cell culture inserts for 24-well plates with pores 0.4 µm in diameter (cellQART^®^, Product #932040). The HFOs were kept in a culture for 3 weeks as described in the Section 2.6. The HFOs were then treated with the indoles and FICZ prior to TNFα challenge for 72 h. A measurement of the trans-epithelial electrical resistance (TEER) was used to characterize the formation of a tight intestinal cell monolayer. TEER electrodes were recorded everyday with an EVOM-resistant meter (WPI, Sarasota, FL, USA), with one electrode placed on the luminal side and the other electrode on the basolateral side, with the two different electrodes only being separated by the intestinal monolayer. The TEER measurements of the filters with the cells were subtracted by blank filters, and the result was multiplied by the membrane area to obtain the TEER measurement in Ω·cm^2^.

### 2.11. Statistical Analysis

All analyses and graphical representations were carried out using GraphPad Prism version 9.1.0 (San Diego, CA, USA). The data were analyzed for normal distribution using the Anderson–Darling test, the d’Agostino and Pearson test, the Shapiro–Wilk test, and the Kolmogorov–Smirnov test. For the data which failed the normality test, nonparametric tests were used to analyze significant differences. Two-group analyses were performed using the two-tailed Student’s *t* test or the nonparametric Mann–Whitney test. The median (interquartile range) and means ± SEM were used to present the data from one of two independent experiments. Multiple comparisons were performed using a one-way analysis of variance (ANOVA) and the post hoc Tukey test or nonparametric Kruskal–Wallis test followed by a post hoc Dunn test. Significance was considered with *p* < 0.05. The different tests are specified in each figure legend.

## 3. Results

### 3.1. Fecal Samples from Preterm Neonates Display Lower Levels of Trp Metabolites Compared to Full-Term Neonates

Our initial objective was to investigate whether there is a differential abundance of Trp metabolites in the feces of preterm neonates compared to full-term neonates. To this end, LC-MS was employed to determine the composition and concentration of Trp metabolites in the feces of a Dutch cohort of preterm (*n* = 19) and full-term (*n* = 9) neonates, at birth and 4 weeks postnatal (Figure 1A). The Trp metabolomic quantification focused on the three metabolic pathways of Trp, namely, the kynurenine, serotonin, and indole pathways (Appendix A). No difference was observed in the fecal concentration of Trp between preterm and full-term neonates at birth and 4 weeks postnatal (Figure 1B). In addition, at birth, no significant differences in Trp metabolites were observed between preterm and full-term neonates if the sum of the metabolites involved in each Trp metabolic pathways was calculated (Figure 1C). The full-term neonates showed higher levels of Trp metabolites in all three Trp metabolic pathways (Figure 1C). Furthermore, at a single-metabolite level, the preterm fecal samples displayed significantly lower levels of xanthurenic acid and serotonin and higher levels of 3-OH-kynurenine (Figure 1D). In the kynurenine pathway, kynurenine was elevated in the preterm neonates, whereas kynurenic acid was elevated in the full-term neonates (Appendix A). At 4 weeks postnatal, the preterm neonates showed no significant difference in the sum of metabolites involved in the kynurenine pathway (Figure 1C). However, at a single-metabolite level, picolinic acid, kynurenic acid, and xanthurenic acid were significantly higher in the feces of the full-term neonates (Figure 1E). Furthermore, at 4 weeks postnatal, the preterm neonates displayed significantly lower fecal levels when regarding the total Trp metabolites involved in the serotonin pathway compared to the full-term neonates (Figure 1C). This difference was mainly explained by a significant reduction in serotonin, *N*-acetyl-serotonin, and 5′HT-OH-indole acetic acid in the feces (Figure 1E). Moreover, the preterm neonates showed significantly lower levels of total Trp metabolites involved in the indole pathway compared to the full-term neonates (Figure 1C). Particularly, the preterm neonates showed significantly lower levels of indole-3-lactic acid (ILA), the main contributor in the sum of indoles in neonates (Figure 1E). Furthermore, the preterm neonates displayed a significantly lower conversion of ILA from Trp in comparison to the full-term neonates, as shown through the ILA-to-tryptophan ratio (Figure 1F). In addition, we confirmed the ability of ILA to activate AHR in a dose-dependent manner, with the concentration of ILA measured in the feces of preterm neonates being too low to activate AHR (Figure 1G). Moreover, when looking at associations between the preterm and full-term metabolites, the ILA-to-indole-3-aldehyde (I3Ald) ratio displayed the highest association signature power linked to full-term neonates (AUC, 0.905, *p* = 4.0644 × 10^−4^, Appendix A). The rest of the measured Trp metabolites were not significantly different (Appendix A).

### 3.2. Human Milk Contains Tryptophan Metabolites

We next profiled the Trp metabolites in human milk (HM, *n* = 22) and commercially available infant milk formula (F, *n* = 3) (Figure 2A). When comparing human milk and formula, no significant difference in the Trp levels was observed (Figure 2B). We then assessed the sum concentrations of all the metabolites contained in each distinct Trp metabolic pathway. No difference was observed between human milk and formula in the sum concentrations of the metabolites from the kynurenine pathway (Figure 2C). However, at a single-metabolite level, picolinic acid was significantly higher while 3OH-anthranilic acid, xanthurenic acid, kynurenic acid, and kynurenine were significantly lower in the formula compared to human milk (Figure 2D). Interestingly, human milk contained significantly higher levels of metabolites from the serotonin pathway compared to formula (Figure 2C). This reflected the relatively high concentration of 5-OH-indole acetic acid in human milk (Figure 2D). All the other measured Trp metabolites were not significantly different (Appendix A). With these data, our next step was to investigate whether the bacterial-derived metabolites were different in the two types of milk. We then assessed the composition and levels of the metabolites representing the indole pathway in human milk and commercial infant formula. The total indole metabolites’ level was significantly higher in the human milk samples compared to the formula (Figure 2C). Specifically, indole-3-acetic acid (IAA) and indole-3-sulfate (I3S), two Trp metabolites known for being AHR agonists, were detected at significantly higher levels in human milk (Figure 2D). 

### 3.3. Neonatal and Milk Indoles Trigger AHR Activation

Here, we investigated the functional relevance of neonatal ILA and human milk IAA and I3S, focusing on their ability to trans-activate the downstream signaling of AHR in epithelial cells. First, we investigated the capability of high (full-term) and low (preterm) ILA levels to activate AHR in a dose-dependent manner using a mouse luciferase reporter cell system. At the concentration detected in the feces of preterm neonates, ILA was not able to activate AHR, while the full-term ILA concentration exhibited a high activation of AHR (Figure 1F and Figure 3A). We then investigated whether the levels of IAA and I3S detected in human milk could similarly trigger AHR activation. Both IAA and I3S significantly activated AHR at the physiological concentrations measured in human milk (Figure 3B).

AHR metabolites exhibit varying affinities for the human AHR receptor compared to their interaction with the mice AHR receptor [31]. Thus, we investigated whether ILA, IAA, and I3S would similarly activate AHRs in human epithelial cells. To visually track the translocation of AHR to the nucleus of human epithelial cells, we challenged Caco2 cells with ILA, IAA, and I3S, at the levels detected in the feces and the milk, and with the synthetic AHR activator FICZ. Notably, in the untreated Caco2 cells, AHR localization was confined exclusively to the cytoplasm (Figure 3C,D), whereas challenge with IAA, I3C, ILA, and FICZ induced the translocation of AHR to the nucleus (Figure 3C,D). Moreover, the nuclear translocation of AHR upon challenge with ILA, IAA, and I3S was fully mitigated with an AHR inhibitor (CH223191) (Figure 3C), confirming the AHR-specific route of action of these indoles.

### 3.4. ILA, IAA, and I3S Reduce IL8 Expression in Caco2 Cells and HFOs through AHR Activation

Having confirmed the ability of IAA, I3S, and ILA to trigger AHR activation at physiological concentrations, along with the subsequent downstream target gene activation (CYP1A1), we then studied the possible role of these AHR-activating metabolites in reducing cytokine-induced epithelial IL8 production in Caco2 and human fetal intestinal organoids (HFOs). In both Caco2 cells and HFOs, TNFα stimulation induced the production of IL8. Guided by findings from our Trp metabolite analysis, we then exposed the Caco2 cells and HFOs to the highest concentrations of AHR activation-inducing indoles observed in the human milk and full-term fecal samples, as described in Figure 4A. In the Caco2 cells, TNFα significantly increased the expression of IL8, which was significantly reduced by IAA, ILA, and FICZ pretreatment (Figure 4B). However, treatment with I3S did not yield such a significant affect (Figure 4B). Upon treatment with CH223191, ILA, IAA, and FICZ lost their ability to reduce the expression of IL8 (Figure 4C). The same results were obtained when measuring the production of IL8 in the supernatant of the cells (Figure 4D). In addition, gene expression of the AHR downstream target gene CYP1A1 was induced in all the conditions, confirming specific AHR activation upon ILA, IAA, I3S, and FICZ challenge (Figure 4E). Upon treatment with CH223191, ILA, IAA, and FICZ lost their ability to activate AHR, shown by a decreased expression of the AHR target gene CYP1A1 (Figure 4E).

To corroborate these findings, we employed human HFOs, a complex in vitro model which more closely resembles the immature intestinal environment of neonates. The HFOs were derived from the intestines of fetuses from 18 to 22 weeks of gestation and then pretreated with ILA, IAA, I3S, and FICZ prior to TNFα challenge (Figure 5A). Upon ILA and IAA treatment, the HFOs produced significantly lower levels of TNFα-induced IL8 protein in the supernatant of the culture (Figure 5B). No difference was evident in TNFα-induced IL8 production after FICZ and I3S pretreatment. Finally, barrier integrity was measured in the HFOs grown two-dimensionally in transwell systems. The HFOs were exposed apically to the IAA and I3S measured in human milk and challenged with TNFα for 72 h. Upon TNFα exposure, the HFOs showed a significant loss in barrier integrity, which was characterized by a reduction in TEER (Figure 5C). The HFOs exposed to ILA and IAA were able to maintain a stable TEER from TNFα-induced barrier loss, whereas I3S showed a significant reduction in barrier integrity (Figure 5C,D and Appendix A).

These data demonstrate that physiological concentrations of IAA and ILA can reduce the production of the neutrophil chemotactic factor IL8 in fetal intestinal epithelial cells and protect barrier integrity.

## 4. Discussion

Tryptophan metabolism in early life has yet to be fully elucidated, particularly in preterm neonates [34,35]. In preterm neonates, the immaturity of metabolic systems and the changes in bacterial colonization and development can significantly impact the intricate balance of tryptophan utilization and conversion [36,37,38]. Here, we have demonstrated that breastfed preterm neonates have a significant differential abundance in fecal tryptophan metabolites compared to breastfed full-term neonates. In addition, we have analyzed the composition and levels of the Trp metabolites in human milk versus formula. In this study, we reported that fecal- and milk-derived metabolites ILA and IAA can reduce the expression of *IL8* and protect barrier integrity in intestinal cells, thus highlighting a deep connection between milk type and microbiota in early life.

We first showed an altered Trp metabolism in all the Trp metabolic pathways (kynurenine, serotonin, and indoles) associated with preterm birth. In the kynurenine pathway, the preterm fecal samples displayed significantly lower levels of xanthurenic acid (XANA) at birth and at 4 weeks of age as well as low levels of kynurenic acid (KYNA) at 4 weeks of age. XANA and KYNA are known metabolites from the kynurenine pathway, generated from the conversion of 3-OH-kynurenine and kynurenine by the enzyme aminoadipate aminotransferase. Recently, XANA and KYNA have been identified to be negatively associated with intestinal inflammation, partially through AHR activation [39]. Interestingly, in our study, the preterm neonate samples displayed higher fecal levels of the KYNA and XANA precursors, namely, 3-OH kynurenine and kynurenine, but lower levels of KYNA and XANA, suggesting an alteration in the host enzyme aminoadipate aminotransferase.

The preterm neonates also exhibited alterations in the indole pathway—particularly, lower levels of Trp-derived AHR-activating agonists, such as ILA. We showed that the preterm neonates converted Trp into I3Ald rather than ILA, suggesting that ILA conversion may be a strong indicator of full-term status and that I3Ald may be a strong indicator of preterm status.

ILA is a known Trp metabolite produced by *Bifidobacterium* and *Lactobacillus* spp., two bacterial species which colonize the intestine of healthy full-term neonates [22,40,41]. ILA has been described to display immunomodulatory properties through the activation of the aryl hydrocarbon receptor (AHR) [22,42]. These results highlight the presence of specific bacteria in full-term neonates compared to preterm neonates. The *Bifidobacterium* species has been shown to convert Trp and indole into ILA [22,42]. For example, a previous study demonstrated that ILA was a metabolite of *B. Infantis*, a bacterium more abundant in full-term neonates [22].

Subsequently, we investigated the consequences of reduced levels of ILA observed in the feces of preterm neonates on the intestinal epithelium. We first confirmed the ability of fecal-derived ILA to activate AHR in epithelial cells. Then, we showed the beneficial properties of fecal-derived ILA on the intestinal epithelium. ILA was able to reduce the expression of IL8 and enhance barrier integrity in an epithelial cell culture after challenging with TNFα, a known inflammatory cytokine which can disrupt epithelial barrier integrity [43]. We confirmed this observation in HFOs, a more complex in vitro intestinal epithelial culture model [44]. The use of an AHR inhibitor in both cell and HFO experiments confirmed the direct role of AHR in these mechanisms. These results demonstrate that preterm birth is associated with an altered Trp metabolism, with a reduction in the metabolites able to activate the immunomodulatory receptor AHR.

We then investigated the potential of human milk nutrition to rewire the Trp metabolism of preterm neonates. Despite extensive studies on the composition and properties of human milk [45,46,47] in recent years, the presence and role of Trp metabolites in human milk remain poorly investigated. We first measured the levels of Trp metabolites in human milk from the Dutch Human Milk Bank and compared them with those measured in infant formula. In the kynurenine pathway, we measured higher levels of XANA and KYNA in the human milk compared to the formula. We suggest that the higher presence of XANA and KYNA in human milk could counterbalance the lower levels of KYNA and XANA measured in the feces of preterm neonates.

In the indole pathway, we detected the presence of two AHR-activating metabolites, namely, I3S and IAA, in human milk. I3S is a metabolite co-produced by human liver cells and the gut microbiota and is a downstream metabolite of the enzymatic reaction of tryptophanase [48]. IAA has been reported to be produced in the gut by bacteria and fungi and is catalyzed by the enzyme tryptophan 2-monooxygenase [49]. Recent studies linked reduced levels of IAA with increased levels of kynurenine in the feces of IBD patients compared to healthy subjects [15], suggesting a beneficial role of IAA in intestinal homeostasis.

We then investigated the beneficial role of milk-derived IAA and I3S on the intestinal epithelium. We first confirmed the ability of milk-derived IAA and I3S to activate AHR in epithelial cells. Then, we showed the beneficial properties of milk-derived IAA and I3S on the intestinal epithelium. Both milk-derived indoles were able to reduce the expression of *IL8* and enhance barrier integrity in epithelial cell and HFO culture systems upon TNFα challenge. These results demonstrate that human milk contains IAA and I3S, two AHR-activating indoles able to maintain intestinal barrier integrity. This is particularly relevant in preterm neonates, in whom a lack of fecal AHR-activating indoles has been observed, as well as an increased susceptibility to developing intestinal failures.

## 5. Conclusions

In summary, our study identified the presence of IAA and I3S in human milk and ILA in the feces of full-term neonates. These AHR metabolites were demonstrated to protect intestinal epithelial barrier function in vitro. However, these findings need to be validated in another cohort, with an increased sample size. Furthermore, we provided insight into how these indoles influence the expression of IL8. Future research should concentrate on elucidating the precise mechanisms through which indoles interact potentially with the NF-κB signaling pathway, thereby offering a deeper understanding of their role in modulating immune responses and inflammation. Moreover, the direct role of altered gut microbiota in preterm neonates on the metabolism of Trp needs to be further investigated using new-generation DNA sequencing. Finally, unveiling the metabolism of Trp in mothers’ milk and the way in which diet could influence both the presence and the absence of specific metabolites would be a crucial step towards a better understanding of the benefits of human milk and could help prevent the development of inflammatory diseases.

## Figures and Tables

**Figure 1 nutrients-16-01531-f001:**
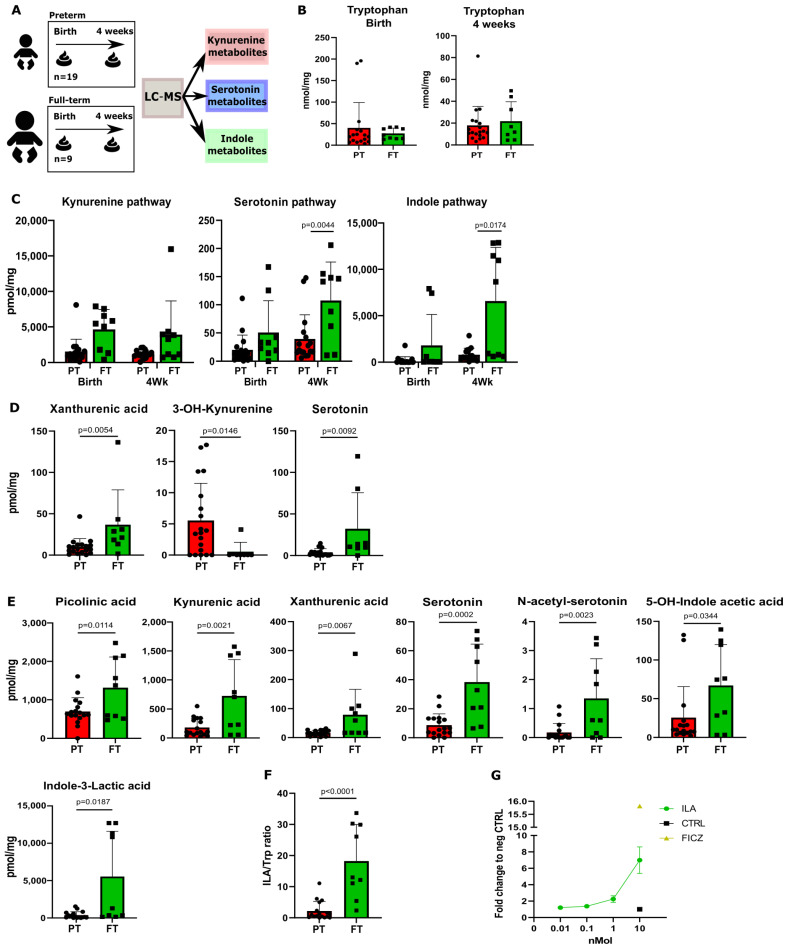
Fecal tryptophan metabolites are highly present in full-term neonates’ feces compared to preterm neonates’ feces. (**A**) A schematic overview of the experimental setup of the tryptophan metabolite analysis performed on the fecal samples of preterm (*n* = 19) and full-term (*n* = 9) neonates. (**B**) Tryptophan levels at birth and after 4 weeks between preterm (PT) and full-term (FT) neonates. (**C**) Total sum of metabolites involved in the kynurenine, serotonin, and indole pathways in fecal samples from preterm and full-term neonates at birth and at 4 weeks of age. (**D**) Metabolite concentrations (pmol/mg) of significantly changed fecal tryptophan metabolites at birth. (**E**) Metabolite concentrations (pmol/mg) of significantly changed fecal tryptophan metabolites after 4 weeks. (**F**) The full-term fecal samples show a higher conversion of tryptophan into ILA (calculated as the ILA-to-tryptophan ratio). (**G**) AHR activity on AHR reporter cell line of ILA at physiological concentration. Data are presented as the median with an interquartile range and error bars extending from minimum to maximum or as the mean ± SEM. Statistical significance is calculated by Student’s two-tailed *t*-test (**B**–**D**,**F**) or the Mann–Whitney test (**E**) based on the distribution of the data.

**Figure 2 nutrients-16-01531-f002:**
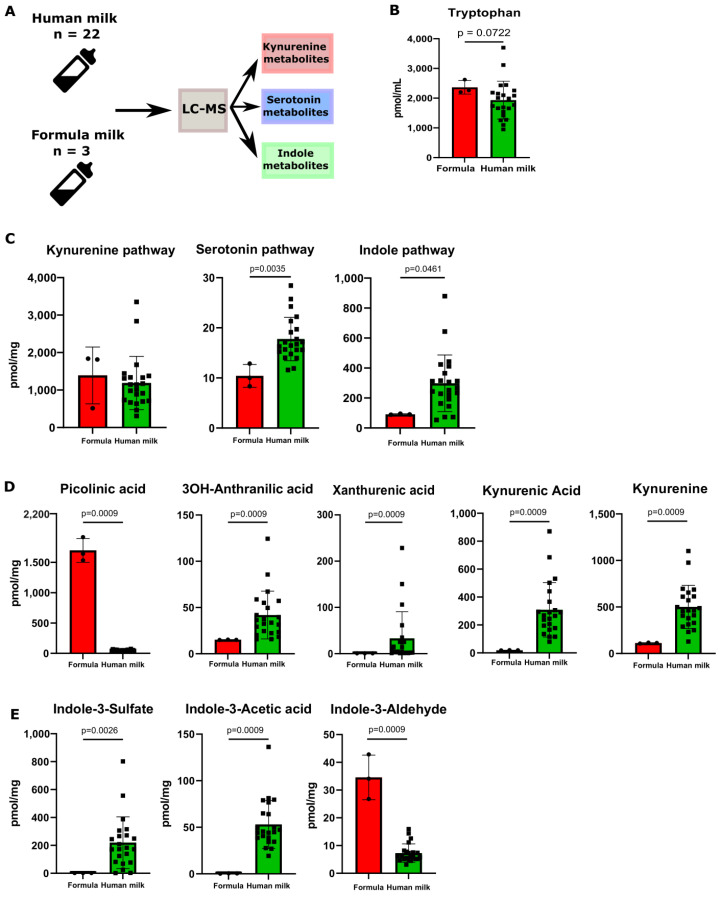
Human milk contains different tryptophan metabolites compared to formula. (**A**) A schematic overview of the experimental setup of the tryptophan metabolite analysis performed on the samples of human milk (*n* = 22) and formula (*n* = 3) is presented above. (**B**) Tryptophan levels between human milk and formula. (**C**) Total sum of metabolites divided between the three metabolic pathways of tryptophan between formula and human milk. (**D**) Metabolite concentrations (pmol/mg) of significantly changed fecal tryptophan metabolites in the kynurenine pathway between human milk and formula. (**E**) Metabolite concentrations (pmol/mg) of significantly changed fecal tryptophan metabolites in the indole pathway between human milk and formula. Data are presented as the median with an interquartile range and error bars extending from minimum to maximum or as the mean ± SEM. Statistical significance has been calculated with the Mann–Whitney test according to the distribution of the data.

**Figure 3 nutrients-16-01531-f003:**
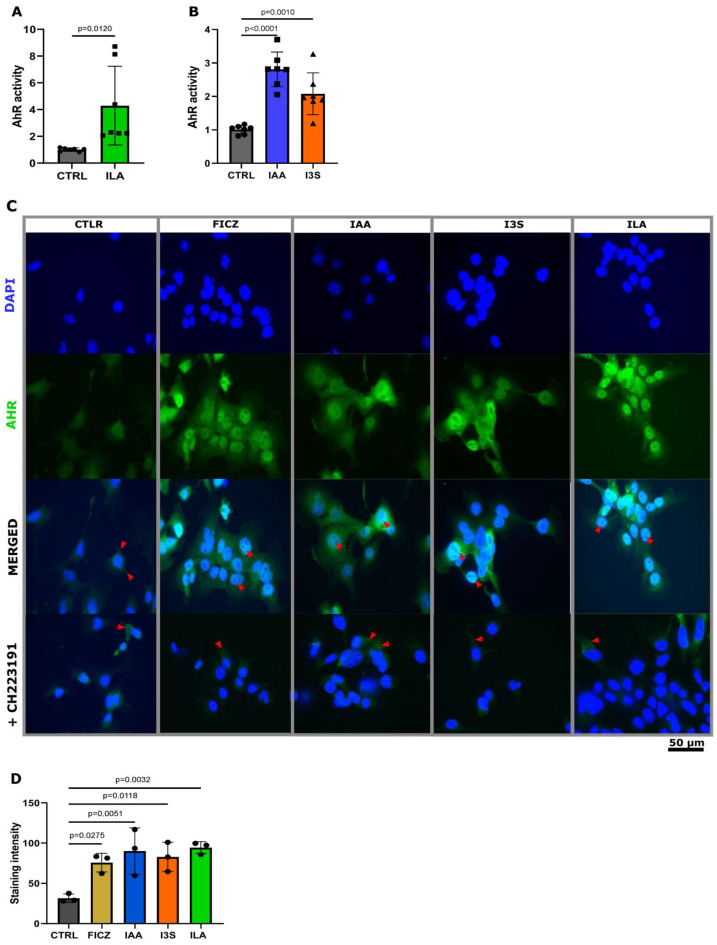
IAA, I3S, and ILA induce AHR translocation into the nucleus of Caco2 cells. (**A**) Quantification of AHR activation of physiological ILA (*n* = 7). (**B**) Quantification of AHR activation of physiological IAA (*n* = 7) and I3S (*n* = 7). (**C**) Staining of nucleus translocation of AHR. Inhibition of translocation with AHR inhibitor CH223191. Red arrow heads are used to indicate translocation into the nucleus. (**D**) Staining intensity was determined using imageJ (ver 1.54f), measuring the mean intensity in the nucleus (*n* = 3). Significant differences between the stainings were evaluated with the Kruskal–Wallis test according to the distribution of the data.

**Figure 4 nutrients-16-01531-f004:**
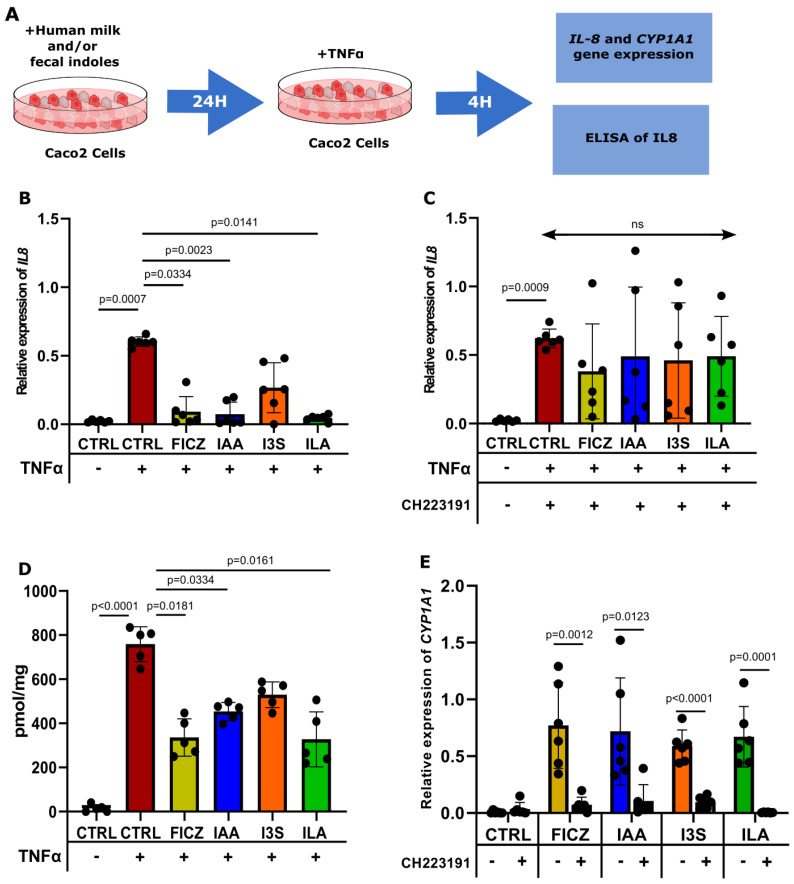
IAA and ILA reduce IL8 expression in Caco2 cells. (**A**) Caco2 cells (*n* = 6), pretreated with the indoles for 24 h, are then challenged with TNFα for 4 h before RNA isolation. (**B**) IL8 expression in Caco2 cells challenged with TNFα. Gene expression has been determined by RT-qPCR (ns = non-significant). (**C**) CH223191 inhibits the reduction in IL8 expression displayed by FICZ, IAA, and ILA. (**D**) IL8 production in the supernatant of Caco2 cells challenged with TNFα. (**E**) CYP1A1 expression in Caco2 cells exposed to the indoles with and without CH223191.

**Figure 5 nutrients-16-01531-f005:**
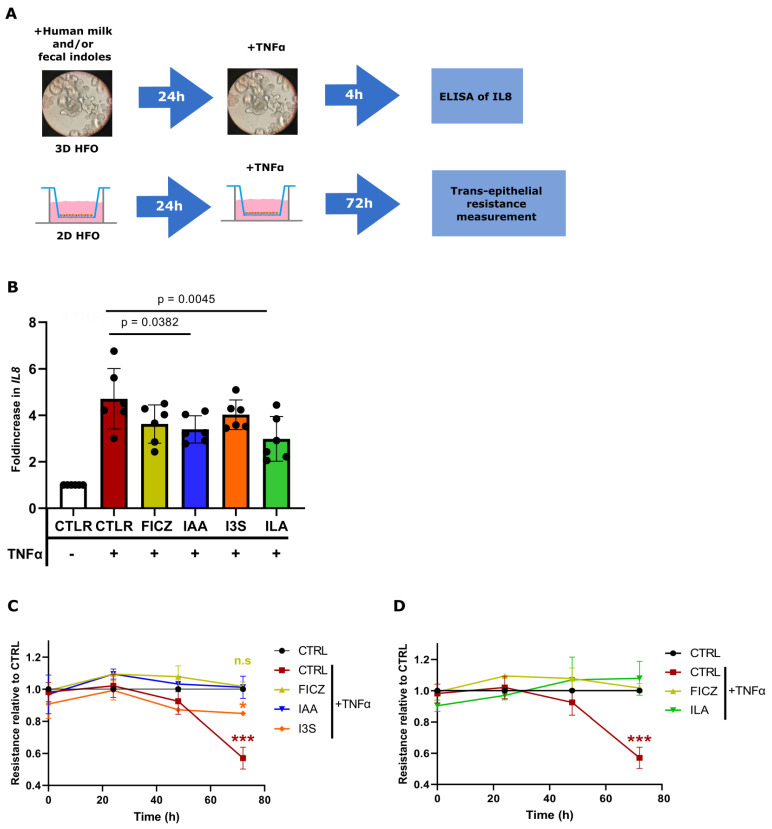
IAA and ILA reduce the production of *IL8* and maintain barrier integrity in HFOs. (**A**) 3D HFOs (*n* = 6) pretreated with human milk and fecal indoles for 24 h and then challenged for 4 h with *TNFα*. IL8 production has been measured through ELISA. 2D HFOs (*n* = 3) pretreated for 24 h with human milk and fecal indoles for 24 h and then challenged with TNFα for 72 h. (**B**) IL8 production in 3D HFOs (*n* = 6, duplicate). (**C**) TEER values of 2D HFOs pretreated with IAA and I3S, relative to the CTRL group. (**D**) TEER values of 2D HFOs pretreated with ILA, relative to the CTRL group. Significant differences between the conditions have been evaluated by a one-way analysis of variance (ANOVA) and a post hoc Tukey test or nonparametric Kruskal–Wallis test, followed by a post hoc Dunn test. Significance is considered with *p* < 0.05 *, *p* < 0.001 ***, n.s = non-significant.

**Table 1 nutrients-16-01531-t001:** Overview of the patient samples included in this study. Values are mean ± SD, *n* (%), or median (interquartile range).

Characteristics	Preterm (*n* = 19)	Term (*n* = 9)
Gestation (W)	27 ± 1.4	>39
Male (%)	10 (56)	4 (44)
Birth weight (g)	1062 ± 272	NA
Vaginal delivery (%)	12 (63)	9 (100)
Breast feeding (%)	19 (100)	8 (88)

## Data Availability

The original contributions presented in the study are included in the article, further inquiries can be directed to the corresponding author.

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
