# Peer review of "Exploring the Immunomodulatory Potential of Human Milk: Aryl Hydrocarbon Receptor Activation and Its Impact on Neonatal Gut Health"

_nutrients, 2024, doi:10.3390/nu16101531_

Round 1

Reviewer 1 Report

Comments and Suggestions for Authors

The results of the present study highlight the importance of tryptophan metabolites through Ahr in the prevention of necrotizing enterocolitis (NEC). Tryptophan is vital for neuronal development, and breast milk is the sole source of the essential amino acid tryptophan. The manuscript is well written, methods are well documented, and figures are presenting convincing results with thoughtful discussion. However, the tryptophan levels shown in the supplementary file should be presented in the main figure which will highlight the importance of the microbiota and their respective individual tryptophan metabolites through the activation of Ahr.

Author Response

We greatly appreciate the reviewers for their valuable time and efforts in reviewing our submission. This letter addresses each issues raised by the reviewers in a point-by-point manner. We have updated the manuscript according to their comments and have modified the figures to improve our submission.

Reviewer 2 Report

Comments and Suggestions for Authors

Investigating the AHR pathway presents several challenges due to its intricate nature. Its complexity stems from multiple factors, including the broad range of ligands that can activate the receptor, the diverse array of target genes it regulates, and its crosstalk with other signaling pathways. My major comments are mainly related to this issue.

Major comments:

How many types of ligands exist for AHRs, and what proportion of them are tryptophan metabolites?

Could you clarify the association between CYP1A1 and IL-8 in the AHR pathway and provide insight into why you have specifically focused on these two downstream molecules of the AHR pathway? What are the main target genes of AHR?

- Unfortunately, the fecal microbiota composition was not assessed in the neonates, which could have provided direct evidence for altered tryptophan metabolism in the gut of preterm neonates.

- The AHR pathway displays species-specific differences. Have differences been described in mice, or is it similar to humans?

Minor comments:

-       Il-8 should be written uniformly throughout (IL-8, IL8, IL-b, I-8)

-       „TNFa@@” -please correct

-       Fig. 4A: beside Caco2, HFO should be also indicated

Author Response

We greatly appreciate the reviewer for their valuable time and efforts in reviewing our submission. This letter addresses each issues raised by the reviewers in a point-by-point manner. We have updated the manuscript according to their comments and have modified the figures to improve our submission.
